

# Left ventricle dysfunction in patients with critical neonatal pulmonary stenosis: echocardiographic predictors. A single-center retrospective study

Carolina D'Anna[1], Alessio Franceschini[1], Micol Rebonato[1], Paolo Ciliberti[1], Claudia Esposito[1], Roberto Formigari[1], Maria Giulia Gagliardi[1], Paolo Guccione[2], Gianfranco Butera[1], Lorenzo Galletti[1] and Marcello Chinali[1]

[1] Department of Cardiac Surgery, Cardiology and Heart and Lung Transplant, Pediatric Hospital Bambino Gesù, Roma, Roma, Italy
[2] Mediterranean Pediatric Cardiology Center "Pediatric Hospital Bambino Gesù", San Vincenzo Hospital, Taormina, Italy

## ABSTRACT

**Background:** The aim of this study is to identify echocardiographic predictors of transient left ventricle dysfunction after pulmonary valve balloon dilatation (PVBD), in neonates with pulmonary valve stenosis (PVS) and atresia with intact septum (PAIVS) at birth.

**Methods:** The study includes patients admitted at the Bambino Gesù Children Hospital from January 2012 to January 2017. Clinical, echocardiographic and cardiac catheterization data before and after PVBD were retrospectively analyzed.

**Results:** Twenty-nine infants were included in the study (21 male and eight female). The median age was $5.8 \pm 7.1$ days. Eight patients developed transient LV dysfunction (three PAIVS and five PVS) and comparing data before and after the procedure, there was no difference in right ventricle geometrical and functional parameters except for evidence of at least moderate pulmonary valve regurgitation after PVBD.

**Conclusion:** Moderate to severe degree pulmonary valve regurgitation was significant associated to LV dysfunction ($p < 0.05$) in PVS and PAIVS patients.

## INTRODUCTION

Pulmonary stenosis (PS) is a congenital heart disease characterized by narrowing of the right ventricle outflow tract (RVOT) and, in most severe forms, complete pulmonary atresia (*Pradat, Francannet & Harris, 2003*). In the latter form, pulmonary blood flow depends completely on patent ductus arteriosus. Pulmonary stenosis (PS) and pulmonary atresia with an intact ventricular septum (PAIVS) represent 25–30% of all cyanotic congenital heart diseases and approximately 3% of all congenital heart diseases. While pulmonary stenosis is relatively common, pulmonary atresia with an intact ventricular

Corresponding author
Carolina D'Anna,
carolina.danna@opbg.net

septum is a complex and uncommon congenital heart defect, associated with a relatively high morbidity and mortality, despite (surgical or percutaneous) treatment. Deaths occurs within the first postoperative year, and survival rates at 15 years of all subtypes of PA-IVS patients is reported to be anywhere from 58% to 87% (*Chikkabyrappa, Loomba & Tretter, 2018*).

The main treatment goal is to achieve anterograde flow across the RVOT in the neonatal period, in order to improve systemic arterial oxygenation.

Trans-catheter intervention with pulmonary valve (PV) balloon dilatation (BD), represents primary therapy for neonates with critical PS and PAIVS. Right ventricular features deemeded necessary to perform PVBD include large, tripartite right ventricle without coronary artery connections and coronary dependent circulation or with interrupted coronary arteries (*Shakeel, 2006*).

It is widely demonstrated that PV perforation and/or BD is characterized by a high rate of initial success with acute gradient reduction, ductus independence and low rate of re-intervention. Nonetheless, significant left ventricle dysfunction has been described in PV/PAIVS after PVBD (*Ronai, 2015*). *Ronai (2015)* described significant transient left ventricular dysfunction after PVBD, as the possible consequence of three hemodynamic effects including the impact of change in size of right ventricle, the loading effect for closure of the ductus arteriosus and the acute change in coronary perfusion.

In previous studies, pre-existing left ventricular dysfunction has been suggested as a risk factor for left ventricle dysfunction after PV perforation and/or BD (*Sholler, Colan & Sanders, 1988*; *Stenberg et al., 1988*; *Gentles et al., 1993*).

The aim of the present study, is to describe the risk factors and hemodynamic mechanisms in patients with PS and/or PAIVS, with normal left ventricle and without coronary anomalies, for the development of left ventricle dysfunction after PVBD.

## METHODS

### Study population

We retrospectively enrolled patients admitted in the Bambino Gesù Children Hospital for neonatal PS and/or PAIVS, from January 2012 to January 2017, with available complete echocardiographic examination, angiograms and operative reports.

Data were retrieved from the clinical records and a dedicated database was collected, after local ethical committee approval (prot 2387_OBPG_2021).

### Management of patients

All patients received intravenous prostaglandin $E_1$ at an initial dose that ranged between 0.005 and 0.1 ng/kg/min.

After clinical stabilization, patients were transferred to the catheterization laboratory. The radiofrequency perforation, when necessary, was performed with 2 F cable with 5 W of energy administered for 1–2 seconds under fluoroscopic guidance. Pulmonary valve was dilated with a balloon measuring 1.2–1.4 times the pulmonary annulus. The procedure was considered effective if the RVOT final gradient was less than 30 mmHg.

Prostaglandin infusion was discontinued 2–5 days after reaching the goal of adequate anterograde pulmonary blood flow (defined as oxygen saturation >92%). In contrast, ductal stent placement or a systemic-to-pulmonary shunt were the treatment options if an additional shunt was required for impossibility to wean patients off from prostaglandin.

## Imaging methods

Imaging data were collected as described in our previous study (*D'Anna et al., 2018*).

Specifically all patients underwent two-dimensional complete echocardiographic examination, using a Philips iE33 (Philips Medical System, Bothell, WA, USA) with 8- and/or 12-MHz transducers before and after pulmonary valve balloon dilatation. Images were analyzed with a dedicated offline review software (Philips Xcelera 4.1 System; Philips Medical System, Bothell, WA, USA).

The echocardiographic views considered were subcostal right and left oblique axis, parasternal long axis, parasternal short axis at the level of ventricle and great arteries, left parasternal (with specific focus on the right ventricle inflow), apical 4- and 5-chambers and suprasternal views. End-diastolic and end-systolic LV volume and RV area and tricuspid Z–score and regurgitation degree were measured in apical 4-chamber view.

Functional parameters included manually traced right ventricular fractional area change (e-RVFAC), speckle tracking automatically derived (STAD) RVFAC by 2D (a-RVFAC), Tricuspid Annular Plane Systolic Excursion (TAPSE), systolic wave of the tricuspid valve on the Tissue Doppler Imaging (TDI) and Right and Left Ventricle 2D Systolic Global Longitudinal Strain (RVGLS and LVGLS). According to the American Society Echocardiography guidelines, all right and left ventricular dimensions were evaluated in end-diastolic phase in both apical and parasternal views.

The Echocardiographic FAC (e-RVFAC) was estimated with the following calculation: [(RVEDA-RVESA/RVEDA) × 100] and the automatic FAC (a-RVFAC) was measured with a speckle tracking automatically derived RVFAC by 2D-STAD using a standard 4-chamber apical view. The Ejection Fraction (LVEF) was estimated with the following calculation: [(LVEDV-LVESV/LVEDV) × 100] Normal RVFAC value was considered greater than 35% (*Lopez et al., 2010*).

TAPSE identified the longitudinal right ventricle function positioning the M-mode cursor on the lateral portion of the tricuspid annulus. Normal value was above 16 mm (*Lopez et al., 2010*).

Tissue Doppler analysis was used to measure systolic tricuspid annulus velocity (Ts'). A velocity below 10 cm/s was considered as a sign of systolic dysfunction.

For strain analysis, cine-loops recordings were reviewed offline and analyzed by single expert operator blinded to diagnosis. Analyses were performed using Philips QLAB software version 10.3 (Philips, Andoven, MA, USA). 2D strain was calculated using apical four-chamber view, focused on the right ventricle with a modified apical view, obtaining a complete image of the right ventricle, particularly the free wall. The speckle tracking strain software developed for the left ventricle was applied also to the right one. RVGLS derived by averaging only strain curves from the RV free wall (4 RV segment model: basal, median, apical free wall and apex). The normal mean values of RVGLS in children were from

**Table 1 Study population demographic features.**

|  | LV dysfunction ($n = 8$) | NO LV dysfunction ($n = 21$) | $p$ |
|---|---|---|---|
| Pre-natal diagnosis | 6/8 (75%) | 7/21 (33%) | NS |
| Sex male | 6/8 (75%) | 15/21 (71%) | NS |
| Mortality | 0 (0%) | 2/21 (9.5%) | NS |
| Days from birth to procedure | 2 ± 1.2 | 8.7 ± 5.2 | 0.04 |
| Gestational age (weeks ± days) | 37.2 ± 1.8 | 37.6 ± 1.8 | 0.58 |
| Height (cm) | 47.6 ± 2.7 | 49 ± 2.6 | 0.19 |
| Weight (g) | 2,807 ± 466 | 3,115 ± 525 | 0.15 |

−20.80% to −34.10% (mean −30.06, 95% CI [−32.91 to −27.21]) and for 10 LVGLS from −16.7% to −23.6% (mean −20.2%, 95% CI [−19.5% to −20.8%]) (*Levy et al., 2014*).

## Statistical analysis

Data are expressed as percentages for categorical data and mean ± standard deviation for normally distributed continuous variables.

Comparison between groups (group 1: patients with left ventricle dysfunction; group 2: patients with normal left ventricle function after PVBD) was carried out by Student's t-test for continuous normally distributed variables and the $\chi^2$ test was used to compare categorical variables.

A $p$ value less than 0.05 was considered statistically significant.

Statistical analysis was performed by SPSS Statistics for Windows software, version 21.0 (released 2012; IBM Corp., Armonk, NY, USA).

## RESULTS

We retrospectively enrolled twenty-nine patients in the study (21 male and eight female). The median age was 5.8 ± 7.1 days. Twenty-five neonates with critical pulmonary valve stenosis (PVS) and four with pulmonary atresia with intact ventricular septum (PAIVS) were included.

Pulmonary valve balloon dilation was performed between one and fourteen days of age. The indications for PVBD were standard for neonates with critical PVS and PAIVS (*Fedderly & Beekman, 1995*).

All patients before the procedure had a normal left ventricle ejection fraction (EF) >57%. After the PVBD, eight patients developed a transient left ventricle dysfunction (five PVS and three PAIVS) with EF <50% calculated by the Simpson's biplane method.

Comparing the demographics and echocardiographic data before the procedure in patients with transient left ventricle dysfunction there was no difference in the demographic data except the timing of PVBD. In fact, in patients who developed LV dysfunction, an earlier PVBD was associated with greater reduction in LV EF (2.0 ± 1.2 days *vs* 8.7 ± 5.2 days, $p = 0.04$) (Table 1).

No significant differences were detected in tricuspid valve annulus diameter Z-score (−0.45 ± 1.3 *vs* −0.72 ± 1.4, $p = 0.839$), tricuspid regurgitation (1.5 ± 0.8 *vs* 1.6 ± 0.8,

**Table 2 Echocardiographic left and right geometrical and functional parameters.**

|  | LV dysfunction ($n = 8$) | NO LV dysfunction ($n = 21$) | $p$ |
|---|---|---|---|
| EF pre-procedure (%) | 64.2 ± 3.6 | 64.2 ± 3.6 | 0.89 |
| EF post-procedure (%) | 41.2 ± 7.1 | 62.5 ± 2.8 | <0.001 |
| Delta EF (%) | −23.1 ± 6.9 | −1.7 ± 2.4 | <0.001 |
| LV STRAIN pre-procedure (%) | −14.1 ± 1.2 | −15.5 ± 1.9 | 0.07 |
| Tricuspid Z-score | −0.45 ± 1.3 | −0.72 ± 1.4 | 0.839 |
| Tricuspid regurgitation | 1.5 ± 0.8 | 1.6 ± 0.8 | 0.321 |
| RV STRAIN pre-procedure (%) | −13.9 ± 3.4 | −14.8 ± 2.5 | 0.192 |

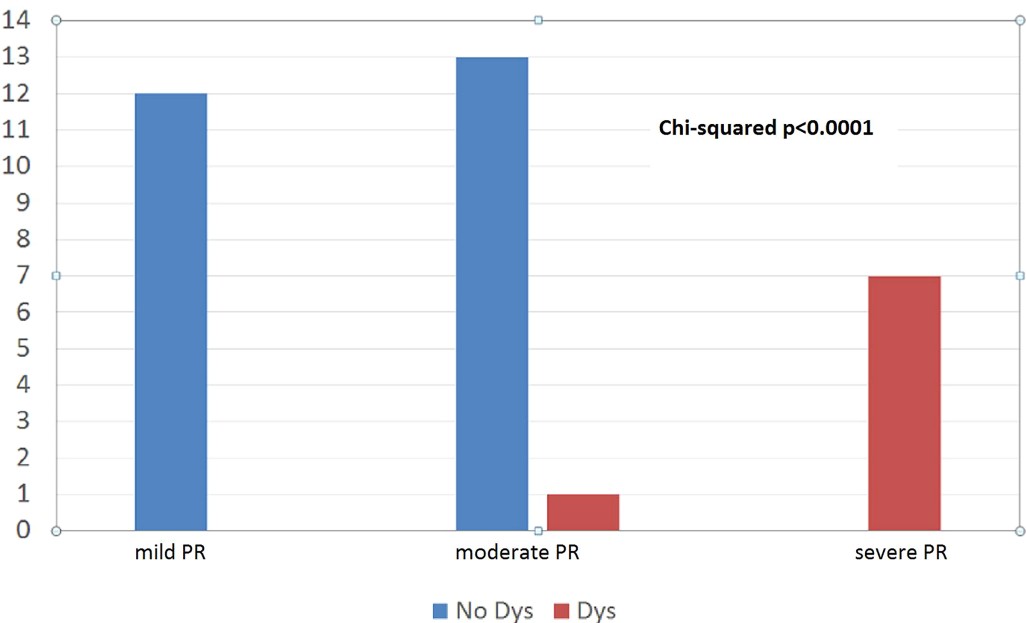

**Figure 1 Moderate pulmonary valve regurgitation after PVBD was significant associated to LV dysfunction ($p < 0.0001$).**

$p = 0.32$) and left (−14.1 ± 1.2% *vs* −15.5 ± 1.9%, $p = 0.07$) and right (−13.9 ± 3.4% *vs* −14.8 ± 2.5%, $p = 0.192$) ventricle longitudinal strain, that appear equally reduced in both groups (Table 2).

However, pulmonary valve regurgitation more than mild degree after PVBD was statistically significant associated with the left ventricle dysfunction ($p < 0.0001$) (Fig. 1).

Patients with a greater change in right ventricle area after PVBD (1.2 ± 0.8 *vs* 0.3 ± 0.9, $p = 0.042$) and left ventricle volume (−1.1 ± 0.9 *vs* −0.1 ± 21, $p = 0.001$) developed left ventricle dysfunction (Table 3).

Ultimately, there was no difference between patients with and without transient left ventricle dysfunction in right and left pressure ratios (RV/LV ratio pre: 1.7 ± 0.4 *vs* 1.5 ± 0.5, $p = 0.122$ and RV/LV ratio post: 0.88 ± 0.18 *vs* 0.80 ± 0.14, $p = 0.350$) (Table 4).

**Table 3 Right and left ventricle geometrical indices before and after pulmonary valve balloon dilatation.**

|  | LV dysfunction (n = 8) | NO LV dysfunction (n = 21) | p |
|---|---|---|---|
| RVD area PRE (cm²) | 2.5 ± 0.8 | 3.0 ± 0.9 | 0.233 |
| RVD area POST (cm²) | 3.7 ± 0.9 | 3.3 ± 0.8 | 0.065 |
| DELTA RVD area (cm²) | 1.2 ± 0.8 | 0.3 ± 0.9 | 0.042 |
| LVED-PRE (mL) | 6.6 ± 2.4 | 6.1 ± 2.0 | 0.538 |
| LVED-POST (mL) | 5.5 ± 2.3 | 6.0 ± 2.1 | 0.030 |
| Delta LVED (mL) | −1.1 ± 0.9 | −0.1 ± 2.1 | 0.001 |

**Note:**
RVD, Right Ventricle Diastolic; LVED, Left Ventricle End-diastolic.

**Table 4 Right and left pressure ratio measured during cardiac catheterization.**

|  | LV dysfunction (n = 8) | NO LV dysfunction (n = 21) | p |
|---|---|---|---|
| RV/LV ratio pre | 1.7 ± 0.4 | 1.5 ± 0.5 | 0.122 |
| RV/LV ratio post | 0.88 ± 0.18 | 0.80 ± 0.14 | 0.350 |

## DISCUSSION

The aim of the study was to investigate incidence and risk factors for the development of left ventricle dysfunction in neonates with PS or PAIVS undergoing PVBD.

Several studies suggest that abnormal intraventricular interaction may be the reason of left ventricle dysfunction with an important prognostic effect on mortality (*Friedberg, 2018*; *Friedberg & Mertens, 2012*; *Santamore & Dell'Italia, 1998*).

Since the right ventricle shares myocardial fibers, interventricular septum and pericardium with the left ventricle, it is intuitive that significant changes in geometry and function of one ventricle involves the contralateral one, independently of neural, humoral or circulatory effect (*Damiano et al., 1991*; *Baker et al., 1998*; *Dexter, 1956*).

*Burkett et al. (2015)* have shown the role of ventricular interdependence in influencing left ventricle function in patients with pulmonary hypertension. A reduced left ventricle longitudinal and circumferential strain and strain rate, primarily at the basal septum, as a consequence of the leftward septal shift of the right ventricle, has been demonstrated in children and young adult with pulmonary hypertension, suggesting direct pressure-loading effects on right and left ventricle performance and hemodynamics (*Burkett et al., 2015*).

The theory of interventricular interaction and its effect on left ventricle function has been demonstrated also in patients with Tetralogy of Fallot both for volume and pressure overload (*Zervan et al., 2009*). Patients with repaired Tetralogy of Fallot (rToF) and pulmonary regurgitation (PR) have a different pathophysiological response to RV chronic volume overload but share with PS and PAIVS indirect worsening of left ventricle function worsening. It has also been shown that patients with rToF and PR have altered RV longitudinal mechanical performance and a tendency to right systolic dysfunction as shown in a previous study from our institution (*D'Anna et al., 2018*), and that the

pulmonary valve replacement in these patients improves global LV strain (*Menting et al., 2015*; *Dragulescu & Friedberg, 2014*). Moreover, in patients with rToF, residual RV outflow tract obstruction induces an early protective effect on RV modeling and RV strain, but a negative impact on LV strain (*Latus, Hachmann & Gummel, 2015*).

As compared to patients with rToF and PR, patients with PVS/PAIVS show a more severe RV diastolic dysfunction with a restrictive physiology, mainly due to RV ventricular hypertrophy, myocardial disarray and fibrosis (*Cheng et al., 2019*).

*Ronai (2019)* colleagues have suggested that, in patients with pulmonary stenosis/atresia after PVBD, worsening of LV longitudinal and circumferential global and segmental strain is a predictor of left ventricle dysfunction, while longitudinal RV strain remains unchanged pre- and post-PVBD. The authors also suggested that right ventricle volume overload was responsible for subsequent left ventricle dysfunction.

The mechanism involved in altering myocardial performance in the ventricular septum is a consequence of the change in RV volume loading conditions, following right ventricular outflow obstruction relief. Patients who developed LV dysfunction after PVBD had larger right ventricles but not significantly larger left ventricles (*Ronai, 2019*).

In our study, left and right ventricular strains before PVBD are reduced, although not statistically significantly, in patients who develop left ventricle dysfunction. In agreement with Ronai's study, the greatest increase of right ventricle area after PVBD in patients with left ventricle dysfunction is statistically significant.

This evidence has confirmed that the right ventricle volume overload (RVVO) after PVBD, due to the iatrogenic development of pulmonary regurgitation, is a predisposing risk factor for transient left ventricle dysfunction.

The underlying mechanism depends on the resultant acute right volume overload, regardless of the reduction of pressure load, which can alter right chamber geometry. This then can cause left ventricle dysfunction due to the known physiological processes of ventricular interdependence and also the decreased relative contribution of left atrial systole to left ventricular filling.

Acute right volume overload induces flattening of the ventricular septum resulting from leftward displacement of the septum toward the center of the left ventricle, opposing the normal forces of left ventricle distension. Normal ventricular septal curvature recovers by the end of systole, which opposes the inward motion of the ventricular septum toward the center of the left ventricle during systole contraction. As a result, net shortening along the ventricular septum-to-posterolateral free wall short axis in RVVO is reduced.

The regional nature of LV impairment depends on ventricular septal flattening, strongly refutes a systemic mechanism explanation (loading alteration, neurohumoral interaction, autonomic influence) and shows that transient left ventricle dysfunction is a consequence of acute RVVO (*Lin et al., 1994*; *Louie et al., 1995*).

In our study, in all patients the greater impairment of longitudinal regional strain in the septum than in the lateral free wall supports the most accredited theory for transient left ventricle dysfunction and demonstrates that it depends on volume overload and interventricular dependence (Tables 5 and 6). We have also considered that the creation of a 'detrimental shunt' could have played a role in the etiology of the LV dysfunction.

**Table 5  Regional septum strain analysis before and after pulmonary valve balloon dilatation.**

| All patients ($n$ = 29) | LV regional strain pre-PVBD | LV regional strain post-PVBD | $p$ |
|---|---|---|---|
| Apical septum | −27.57 | −16.11 | <0.04 |
| Medium septum | −11.54 | −7.62 | 0.1 |
| Basal septum | −10.5 | −7.6 | <0.001 |

**Table 6  Regional left ventricle lateral wall before and after pulmonary valve balloon dilatation.**

| All patients ($n$ = 29) | LV regional strain pre-PVBD | LV regional strain post-PVBD | $p$ |
|---|---|---|---|
| Lateral-apical | −20.68 | −15.6 | 0.34 |
| Lateral-medium | −18.14 | −16.17 | 0.37 |
| Lateral-basal | −12.48 | −9.52 | 0.28 |

**Table 7  Left ventricle dysfunction patients main features.**

| Patients $n$ | Patent ductus | Flow direction (left-right) | PGE stop (days) | Ductus closure (days) | Time to recovery EF (days) | Pulmonary regurgitation degree before EF recovery | Pulmonary regurgitation degree after EF recovery |
|---|---|---|---|---|---|---|---|
| 1 | Yes | Yes | 3 | 20 | 20 | Severe | Mild |
| 2 | Yes | Yes | 3 | 20 | 120 | Severe | Moderate |
| 3 | Yes | Yes | 3 | 7 | 6 | Severe | Mild |
| 4 | Yes | Yes | 2 | 40 | 150 | Severe | Moderate |
| 5 | Yes | Yes | 8 | 10 | 13 | Severe | Moderate |
| 6 | Yes | Yes | 23 | 25 | 25 | Severe | Mild |
| 7 | Yes | Yes | 2 | 2 | 20 | Moderate | Mild |
| 8 | Yes | Yes | 3 | 3 | 7 | Severe | Mild |

In the presence of a severe pulmonary regurgitation, a detrimental shunt due to the wide patent ductus arteriosus can be present. Because of the significant pulmonary regurgitation, blood coming into the pulmonary artery by the wide ductus arteriosus is "drawn" to the right ventricle leading to systemic and coronary stealing. The establishment of this detrimental shunt is sometimes terms as "circular shunt" in the setting of congenital heart disease (*Hakim et al., 2013*). However, the theory of the detrimental circular shunt is unsupported by our data. In fact, we found no relation between ductus closure and left ventricle dysfunction improvement. Furthermore, our data show that LV dysfunction is mainly related to a segmental motion abnormality rather than global dysfunction (as would be expected in reduced coronary flow) and to the degree of pulmonary regurgitation (Table 7).

## Limitations

This is a single-center retrospective study with a small sample size, even for the low percentage of cases due to the rarity of this congenital heart disease. Potential selection biases include the influence of multiple factors such as heart rate, preload and afterload in

the echocardiographic evaluation of functional and morphological indices. Studies with a larger sample size and future collaborations with other centers are therefore needed to demonstrate the universality of this observation.

## CONCLUSION

Moderate-severe degree pulmonary valve regurgitation predisposes to transient left ventricle dysfunction in patients with PVS and PAIVS after PVBD. The acute right ventricle volume overload, which also influences the left ventricle through the known pathophysiological mechanisms of ventricular interdependence, is the implied mechanism. This hypothesis could be a starting point for defining features of patients at higher risk to develop ventricular dysfunction and to draw up a protocol for greater surveillance for improving clinical management to ensure a better outcome for these patients.

### Funding
The authors received no funding for this work.

### Competing Interests
The authors declare that they have no competing interests.

### Author Contributions
- Carolina D'Anna conceived and designed the experiments, performed the experiments, analyzed the data, prepared figures and/or tables, authored or reviewed drafts of the article, and approved the final draft.
- Alessio Franceschini conceived and designed the experiments, performed the experiments, analyzed the data, prepared figures and/or tables, authored or reviewed drafts of the article, and approved the final draft.
- Micol Rebonato conceived and designed the experiments, performed the experiments, analyzed the data, prepared figures and/or tables, and approved the final draft.
- Paolo Ciliberti conceived and designed the experiments, performed the experiments, analyzed the data, prepared figures and/or tables, authored or reviewed drafts of the article, and approved the final draft.
- Claudia Esposito conceived and designed the experiments, performed the experiments, analyzed the data, prepared figures and/or tables, and approved the final draft.
- Roberto Formigari conceived and designed the experiments, performed the experiments, analyzed the data, prepared figures and/or tables, analysis tools, and approved the final draft.
- Maria Giulia Gagliardi conceived and designed the experiments, performed the experiments, analyzed the data, prepared figures and/or tables, analysis tools, and approved the final draft.

- Paolo Guccione conceived and designed the experiments, performed the experiments, analyzed the data, prepared figures and/or tables, authored or reviewed drafts of the article, analysis tools, and approved the final draft.
- Gianfranco Butera analyzed the data, authored or reviewed drafts of the article, and approved the final draft.
- Lorenzo Galletti analyzed the data, authored or reviewed drafts of the article, and approved the final draft.
- Marcello Chinali conceived and designed the experiments, performed the experiments, analyzed the data, prepared figures and/or tables, authored or reviewed drafts of the article, and approved the final draft.

## Data Availability

The raw data is available in the Supplemental Files.

## Supplemental Information

Supplemental information for this article can be found online at http://dx.doi.org/10.7717/peerj.14056#supplemental-information.

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
