# Peer review of "Left ventricle dysfunction in patients with critical neonatal pulmonary stenosis: echocardiographic predictors. A single-center retrospective study"

_PeerJ, doi:10.7717/peerj.14056_

## Round 0.1 · original submission · Major Revisions

I have included an annotated PDF highlighting some of the many places that should be improved in your revision, if you decide to resubmit. The enclosed review makes other points that would all need to also be taken into account.

Aside from the lack of clarity in communication, for which the errors in English grammar and use can be addressed by having your manuscripted edited by an appropriate service or a native English-speaking colleague, the presentation of your single figure should be improved.

More importantly, as you yourselves have pointed out, "the conclusions may be underpowered [making it] difficult to compare outcomes with other studies... a larger sample size [is] needed." If you choose to resubmit, you must include the justifications of and calculations for a power analysis. You may find that the conclusion is to enlarge your sample of patients, at which point I encourage you to try to make your study multicentric in collaboration with another hospital. If you need more time before resubmission in this specific circumstance, please contact us and we can arrange it. Thank you for your submission.

Reviewer 1 ·

Basic reporting

This entire manuscript is very hard to understand and follow the reasoning and hypothesis. There should be numbers cited for how common or rare the disease is and the to support the comments about mortality. The literature review is comprehensive and appropriate

Experimental design

The numbers are small but they raise an interesting hypothesis.

Validity of the findings

Table 1 is unnecessary - the point was to see a difference in the groups between no LV dysfunction and LV dysfunction - and then try to ascertain why that difference is there. Comparing LV dysfunction to no LV dysfunction is not helpful. There should be a standard table 1 - that includes basic demographics (gestational age, weight, sex, prenatal diagnosis, mortality etc).
The circular shunt description while interesting is very poorly worded. It would be helpful to have in table format for the LV dysfunction cases if the ductus was still patent and what direction blood flow was going.
The fact that this dysfunction is transient should also be highlighted - moreover, I would urge the authors to include how long it took for function to normalize, and was there a change in the degree of pulmonary regurgitation on that normal LV function echo.

Additional comments

This is an interesting study that should be published but there are many changes that need to be made to improve the manuscript.

---

## Round 0.2 · Minor Revisions

The revised manuscript communicates the authors' message much more clearly and is now acceptable for publication. Please find attached some copy-level editing suggestions to polish the last rough edges. Highlighted text is to be removed or corrected, blue is to add in or replace non-standard wording.

---

## Round 0.3 · accepted · Accept

Thank you for your rapid turn-around and have an excellent end of the summer. I have requested a few last copy-editing changes as follows but am happy to accept the manuscript at this point.

line 36: "survival rates in 15 years" (to "at 15 years")
line 60: "dedicate" (to "dedicated")
line 131: "Not statistically significant differences were detected about" (to "No significant differences were detected in")
line 160: "indirect worsening effect of" (remove "effect")
line 186 This than causes (to "then")